# Immunomodulatory PEG-CRGD Hydrogels Promote Chondrogenic Differentiation of PBMSCs

**DOI:** 10.3390/pharmaceutics14122622

**Published:** 2022-11-28

**Authors:** Meng Yang, Rong-Hui Deng, Fu-Zhen Yuan, Ji-Ying Zhang, Zi-Ning Zhang, You-Rong Chen, Jia-Kuo Yu

**Affiliations:** 1Department of Sports Medicine, Beijing Key Laboratory of Sports Injuries, Peking University Third Hospital, Beijing 100191, China; 2Institute of Sports Medicine, Peking University, Beijing 100191, China

**Keywords:** peripheral blood mesenchymal stem cells (PBMSCs), polyethylene glycol hydrogels, adhesion peptide (CRGD), cartilage repair, macrophages

## Abstract

Cartilage damage is a common injury. Currently, tissue engineering scaffolds with composite seed cells have emerged as a promising approach for cartilage repair. Polyethylene glycol (PEG) hydrogels are attractive tissue engineering scaffold materials as they have high water absorption capacity as well as nontoxic and nutrient transport properties. However, PEG is fundamentally bio-inert and lacks intrinsic cell adhesion capability, which is critical for the maintenance of cell function. Cell adhesion peptides are usually added to improve the cell adhesion capability of PEG-based hydrogels. The suitable cell adhesion peptide can not only improve cell adhesion capability, but also promote chondrogenesis and regulate the immune microenvironment. To improve the interactions between cells and PEG hydrogels, we designed cysteine-arginine-glycine-aspartic acid (CRGD), a cell adhesion peptide covalently cross-linked with PEG hydrogels by a Michael addition reaction, and explored the tissue-engineering hydrogels with immunomodulatory effects and promoted chondrogenic differentiation of mesenchymal stem cells (MSCs). The results indicated that CRGD improved the interaction between peripheral blood mesenchymal stem cells (PBMSCs) and PEG hydrogels. PEG hydrogels modified with 1 mM CRGD had the optimal capacity to promote chondrogenic differentiation, and CRGD could induce macrophage polarization towards the M2 phenotype to promote tissue regeneration and repair. PEG-CRGD hydrogels combined with PBMSCs have the potential to be suitable scaffolds for cartilage tissue engineering.

## 1. Introduction

The injury of articular cartilage caused by trauma and arthritic diseases is one of the leading causes of disability in the aged population [1]. Due to the lack of blood vessels and nerve tissues, once cartilage is damaged, it is difficult to repair by itself [2]. Currently, tissue engineering scaffolds with composite seed cells have emerged as a promising approach for cartilage repair [3]. To achieve viable and functional repair, the biomaterial scaffold needs to have extraordinary properties, including good biocompatibility, non-toxic degradation products, and regulation of the immune microenvironment after injury [4,5]. Therefore, hydrogels are widely used as scaffold materials in tissue engineering due to their high water content and biocompatibility [6,7]. At present, the most commonly explored hydrogel materials for osteochondral tissue engineering include polyethylene glycol (PEG) [7,8], chitosan [9,10], hyaluronic acid [11], chondroitin sulfate [11], and gelatin [12,13]. Among them, synthetic polymers (such as PEG hydrogels) are widely used in tissue engineering due to their characteristics of high hydrophilicity, easy to precise regulation, and low immunogenicity [7,8]. However, PEG is fundamentally bio-inert and lacks the intrinsic cell adhesion capability, which is critical for the maintenance of cell function [14]. Cell adhesion peptides or other polymers are usually added to improve the cell adhesive properties of PEG-based hydrogels [15,16,17]. For example, adhesion domains (fibrinogen-derived arginine-glycine aspartic acid (RGD)) sequences have been covalently coupled with PEG to promote mesenchymal stem cell viability in PEG-based hydrogels.

While RGD has been generally accepted as a peptide to enhance cell viability, its effect on cell differentiation is controversial [18]. Elisseeff et al. found that RGD-conjugated PEG hydrogels could promote osteogenesis of bone marrow mesenchymal stem cells (BMMSCs) and chondrogenesis of human embryonic stem cells (hESCs) through photopolymerization [19,20]. However, Anseth et al. showed that RGD was required for the initial chondrogenesis of human mesenchymal stem cells (hMSCs), but at a high concentration of 10 mM, the persistence of RGD would inhibit chondrogenesis [21,22]. Contrary to the results of Kim [23] and Vonwil [24] that RGD promoted chondrogenesis, Smith showed that with the increase of RGD concentration, the chondrogenic phenotype and extracellular matrix secretion of human chondrocytes was inhibited [25]. Therefore, further studies are needed to investigate the role of RGD in stem cell differentiation.

Another important consideration for using PEG-CRGD hydrogels as scaffolds is their potential cytotoxicity, which may be caused by the chemicals used to prepare the hydrogels. For example, photopolymerization is the most commonly used method for chemically cross-linked PEG-based hydrogels [26,27]. The main drawback of photopolymerization is that gelation involves toxic initiators and ultraviolet light, which could damage the cells of the body. Jeffery explored Michael addition chemical reactions to prepare and synthesize PEG-based hydrogels in physiological environments [28]. It is important to note that this reaction does not require any catalyst or initiator. Therefore, it is a promising method for cell encapsulation in tissue engineering hydrogels. In this study, in order to ensure rapid gelation, an efficient and initiator-free mercaptan maleimide (SH-MAL) Michael addition reaction was selected as cross-linking. The cell adhesion peptide CRGD (cysteine-arginine-glycine-aspartic acid) also contains SH groups and can react with PEG-MAL with Michael addition.

With the further understanding of immune response, researchers found that the polarization of macrophages had a crucially influence on tissue repair processes. Macrophages can be activated as classically activated macrophage (M1), which has a pro-inflammatory effect. They can also be activated as M2, namely, alternatively activated macrophage, which plays an anti-inflammatory role [29] and secretes some growth factors to promote tissue repair [30,31]. Therefore, regulating the local tissue microenvironment by intervening the polarization state of macrophages is a critical role to alleviate early inflammatory response and carry out cartilage damage repair. As biomaterials are foreign substances and most of them are artificial products, they have strong immunogenicity. Therefore, they easily lead to the persistent activation of M1 macrophages, resulting in chronic inflammation and fibrosis, which is not conducive to the survival of biomaterial-loaded MSCs and significantly affects the tissue repair [32]. Studies have shown that RGD can regulate macrophage function by binding specific integrin sites [5]. Therefore, in this study, we explored the immune regulatory function of PEG hydrogels modified with gradient concentrations of CRGD.

The purpose of this study was to synthesize PEG-CRGD polypeptide hydrogels by Michael addition reaction, and to study the effect of CRGD content on PBMSCs’ cartilage formation and macrophage immune regulation. The PEG hydrogels were characterized by scanning electron microscopy, and the effects of CRGD on PBMSCs cell adhesion and cell viability were evaluated by confocal and CCK8 experiments.

## 2. Materials and Methods

### 2.1. Cell Culture

Mouse macrophage cells (Raw264.7) were purchased from the Shanghai Cell Center (Shanghai, China). PBMSCs were isolated from the central artery of the ear from 3-month-old New Zealand white rabbits. Mobilization, isolation, culture, and trilineage differentiation capacity assays were performed as previously described [33]. PBMSCs in the third passage were seeded into hydrogel scaffolds (1 × 10^6^ cells per scaffold).

### 2.2. Preparation of Hydrogels

CRGD was synthesized by solid-phase peptide synthesis (SPPS) method. Briefly, 1 g Trp(Boc)-2-chlorotrityl resin and 10 mL *N*,*N*-dimethylformamide (DMF) were added to the reaction vessel, and the resin was swelled in DMF for 15 min. Dichloromethane (DCM, 18 mL), methanol (3 mL), and *N*,*N*-diisopropylethylamine (DIEA, 1.0 mL) were added for 30 min to seal the unreacted functional groups in the resin. Then, Fmoc-protected amino acids (3.0 mmol), DIEA (1.0 mL), and DMF (8 mL) were added. The reaction was carried out for 1 h under nitrogen flow. After each coupling step, excess reactants were washed with DMF, followed by the addition of DMF (16 mL) and piperidine (4 mL) to remove Fmoc protection. After the last amino acid coupling was completed, the peptide was cut off from the resin by incubating with trifluoroacetic acid (TFA) (26.4 mL), dithiothreitol (DTT, 5 mg), water (5 mL), and triisopropyl silane (2 mL) for 2 h, and the crude product was obtained by extraction and filtration, which was further purified by precipitation with ethyl diether. The molecular weight of the peptide was analyzed by electrospray ionization mass spectrometry (ESI-MS): *m*/*z* ([M+H]^+^) = 450.2 (Calculated); 225.7, 450.3 (Found).

We used 4 hydrogel samples with different CRGD contents (0, 0.5, 1, 2.5 mM). Solutions of polyethylene glycol-sulfhydryl (PEG-SH, 100 mg/mL) and polyethylene glycol-maleimide (PEG-MAL, 100 mg/mL) were mixed with each other (50 µL PEG-SH: 50 µL PEG-Mal) in room temperature to obtain the PEG hydrogel (Figure 1 and Figure 2A). A total of 0.5 mM CRGD-PEG hydrogel was prepared by mixing with 48.87 µL PEG-SH, 50 µL PEG-MAL, and 1.13 µL CRGD; 1 mM CRGD-PEG hydrogel was prepared and mixed with 47.75 µL PEG-SH, 50 µL PEG-MAL, and 2.25 µL CRGD; 2.5 mM CRGD-PEG hydrogel was prepared and mixed with 44.37 µL PEG-SH, 50 µL PEG-MAL, and 5.63 µL CRGD (total mercaptan: maleimide = 1:1) to make a hydrogel with 10% solid content of 100 µL. Gelation occurred after standing at room temperature for 1 min.

### 2.3. Scanning Electron Microscopy

The freeze-dried hydrogels were bonded to the sample table, and the scaffolds were sprayed with gold for 60 s before the JSM-7900F scanning electron microscope (SEM, Hitachi, Japan) observations. SEM images were acquired at an accelerating voltage of 3.0 kV, a working distance of ~14.0 mm, and 500× magnification. The average pore size was determined from SEM images by Image *J* software.

### 2.4. Rheology Measurement of PEG-CRGD Hydrogels

The theology oscillation frequency sweep was conducted by a rotational rheometer (Thermo, Waltham, MA, USA) with an 8 mm parallel probe. The gelation of each hydrogel was taken in a mold with a thickness of 3 mm and then cut into a circle of 8 mm in diameter. The sweep was taken at 25 °C and γ = 0.01 rad/s.

### 2.5. Compressive Testing of PEG-CRGD Hydrogels

Hydrogels were prepared in a mold with 8 mm in diameter and 8 mm in thickness. The compressive test was performed by a material testing machine of Instron 3365 (Instron, Norwood, MA, USA) with a crosshead speed of 3 mm/min at 25 °C.

### 2.6. Swelling Properties of PEG-CRGD Hydrogels

We prepared the hydrogel samples (10 wt%, 50 µL) and recorded the initial weight (*W_d_*), then added 5 mL PBS (pH 7.4) and incubated them at 37 °C. The samples were weighed again (*W_s_*) after incubation for certain time (0.5, 2, 6, 10, 14, 20, and 26 h). The calculational equation of swelling ratio was used as shown below:$$SR=\left(W_{s}-W_{d}\right)/W_{d}$$

### 2.7. Cell Survival and Proliferation in PEG-CRGD Hydrogels

Live/dead cell staining was used to determine the survival and proliferation of PBMSCs in hydrogels. After incubation for 7 days, the hydrogels containing the cells were washed three times with PBS. Then, the PBMSC–hydrogels were immersed in 1 mL of working reagents containing calcein AM (2 μM) and PI (8 μM) at 37 °C for 1 h. The distribution of live (green fluorescence) and dead (red fluorescence) cells was observed under the confocal microscopy (excitation: 488 or 568 nm).

The proliferation of PBMSCs in hydrogels was tested by Cell Counting Kit-8 assay (CCK8). PBMSCs were seeded into 48-well microplates (5 × 10^3^ cells/50 μL/well). After incubation for 1, 3, and 7 days, fresh culture media (500 μL) with CCK8 reagent (50 μL) were added. Then, after 2 h incubation, 100 μL of liquid from 48-well microplates was transferred into a 96-well microplate for recording the optical density (OD) value of 450 nm using a microplate reader (Thermo, Waltham, MA, USA). 

### 2.8. Cell Adhesion on Hydrogels

The hydrogel (40 μL) was spread on the bottom of the 96-well plate, and then the cell suspension was uniformly dropped onto the surface of the hydrogel. After 12 h, the medium was removed and fixed with 4% paraformaldehyde for 30 min, permeabilized with 1% triton X-100 for 30 min, and the F-actin was stained with rhodamine phalloidin (Cytoskeleton, Inc, St. Denver, CO, USA). Then, the fluorescent dye 4′-6-Diamidino-2-phenylindole (DAPI, Sigma, St. Louis, MO, USA) was used as a nuclear stain. Confocal images were taken using the Leica SP8 confocal microscope.

### 2.9. Chondrogenic in PEG-CRGD Hydrogels

For immunofluorescence, after 3D chondrogenic differentiation in hydrogels for 14 days, the PBMSCs cultured in hydrogels were fixed with 4% paraformaldehyde for 30 min. Samples were blocked with 10% bovine serum albumin (BSA, Gibco, Waltham, MA, USA) for 1 h following permeation. Then, rabbit primary anti-collagen II (1:500 dilution, Abcam, Cambridge, UK), 488 goat anti-rabbit IgG antibody (1:1000 dilution, Invitrogen, Waltham, MA, USA) and DAPI were successively incubated with the specimens, and the immunofluorescence intensity of collagen II was investigated by quantifying the immunofluorescence intensities from confocal micrographs.

### 2.10. Reverse Transcription Quantitative Polymerase Chain Reaction (qRT-PCR)

Total RNA was extracted with TRIzol reagent (Invitrogen, USA) from cells cultured on hydrogel biomaterials. RevertAid First Strand cDNA Synthesis Kit (Thermo Scientific, Waltham, MA, USA) was used to reverse transcribe total RNA to cDNA. Gene expression was assessed by a quantitative real-time polymerase chain reaction (qRT-PCR) using real-time PCR system (Applied Biosystems, Waltham, MA, USA). GAPDH was used as an internal control. The relative expression of marker genes was quantified using 2^−∆∆Ct^ method.

### 2.11. Macroscopic and Histological Analysis

For subcutaneous hydrogel implantation of C57BL/6 mice, Histological analysis was performed by using hematoxylin-eosin (H&E, Solarbio, Beijing, China) and Masson’s trichrome (M&T, Solarbio, Beijing, China) staining. Immunohistochemical staining of CD86 (1:500 dilution, Cell Signaling, Danvers, MA, USA) and CD206 (1:500 dilution, Cell Signaling, Danvers, MA, USA) were performed to assess the phenotype of macrophages infiltrated by tissues surrounding the hydrogels after 30 days of implantation. The slices were heat-treated with antigen retrieval solution (Tris/EDTA, pH 9.0) at 100 °C for 20 min.

### 2.12. Statistical Analysis

Each experiment was repeated at least in triplicate. Statistical analysis among groups was conducted using one-way ANOVA with Tukey’s post hoc test or two-way ANOVA with Tukey’s post hoc test. All data analyses were calculated using SPSS 22.0 software (International Business Machines Corporation, USA). The value of *p* < 0.05 was considered as statistically significant.

## 3. Results and Discussions

### 3.1. Morphological Observation and Characterization of Hydrogels

The chemical structures of 4Arm-PEG-SH, 4Arm-PEG-Mal, and CRGD schematic diagrams of various cross-linked reactions in hydrogels were illustrated in Figure 1 and Figure 2A. As shown in the Figure 2B, it can be observed that with the increase of CRGD concentration in PEG hydrogels, the overall morphology of hydrogels changes correspondingly. PEG hydrogels without CRGD have a regular appearance and compact structure. With the increase of CRGD concentration, hydrogels gradually lose their structure. Scanning electron microscopy was used to further observe the pore size of the freeze-dried PEG-CRGD hydrogel (Figure 2C,D). It was found that when the CRGD concentration changed from 0 mM to 1 mM, the average pore size of the hydrogel increased slightly, which may be related to the fact that the introduction of CRGD changed the regular structure of the pure four-arm PEG hydrogels. However, when the concentration of CRGD increased to 2.5 mm, the average pore size decreased slightly. Considering that the gel phase diagram combined with PEG-2.5 mM CRGD may be too high with the concentration of CRGD, the four-arm PEG cannot be properly combined effectively, and some hydrogels collapse, resulting in a slight decrease in pore size.

With the increase of CRGD concentration, the storage modulus G′ of hydrogels gradually decreased in the frequency sweep rheology study (Figure 2E). The G′ of PEG-0 mM CRGD was ~3.6 kPa. PEG-0.5 mM CRGD and PEG-1 mM CRGD showed a comparable G′ of ~2.5 kPa, while the G′ of PEG-2.5 mM CRGD was ~1.3 kPa. The anti-compression performance of the cartilage-repairing hydrogels is especially critical because the cartilage needs to withstand continuous compressive stress. The compression test showed that the PEG-CRGD hydrogels were able to withstand a compressive strain of more than 90% (Figure 2F). The mechanical properties of PEG-CRGD hydrogels decreased with the increase of CRGD concentration. The mechanical properties of PEG-2.5 mM CRGD hydrogels were statistically poorer compared with other hydrogels. The equilibrium swelling ratios of PEG-CRGD hydrogels were found to be 500~600% without statistical significance (Figure 2G).

### 3.2. Biocompatibility of MSCs on Hydrogel Scaffolds

The PBMSCs in hydrogels of each group were observed after 7 days of culturing (Figure 3A). In the fluorescence images, live cells were stained green, while dead cells were stained red. There were more dead PBMSCs in the PEG-0 mM CRGD hydrogel group, which may be related to cell apoptosis due to cell failure to adhere to the hydrogel scaffold. PBMSCs in the PEG-0.5 mM CRGD hydrogel group also had some dead cells, but the number was lower than that in the PEG-0 mM CRGD hydrogel group. PBMSCs in the PEG-1 mM CRGD hydrogel group and the PEG-2.5 mM CRGD hydrogel group had almost no dead cells, and the number of living cells increased significantly compared with the first two groups.

The cell proliferation activity was measured by a CCK8 experiment. Figure 3B showed that PBMSCs in the PEG-0 mM CRGD hydrogel group had the weakest proliferation ability (*p* < 0.05). There was no significant difference in cell proliferation rate between the PEG-1 mM CRGD hydrogel group and the PEG-2.5 mM CRGD hydrogel group at days 1, 3, and 7 (*p* > 0.05). There was no significant difference in cell proliferation rate between the PEG-0 mM CRGD hydrogel group and the PEG-0.5 mM CRGD hydrogel group on day 3 (*p* > 0.05). In conclusion, the proliferation ability of PBMSCs in the PEG-0 mM CRGD hydrogel group was the weakest (*p* < 0.05), and the proliferation capacity of PBMSCs in the PEG-0.5 mM CRGD hydrogel group was increased on day 1 and day 7 (*p* < 0.05). The proliferation ability of PBMSCs was the highest in the PEG-1 mM CRGD hydrogel group and the PEG-2.5 mM CRGD hydrogel group (*p* < 0.05).

### 3.3. The Effect of Gradient Concentration of RGD on Cell Adhesion

A confocal special 96-well plate was added with 40 μL of hydrogels mixed with different concentrations of CRGD. A total of 1 × 10^3^ cells were inoculated on the surface of the hydrogels and cultured for 12 h by cell counting.

Cell adhesion peptides can promote cell adhesion. In order to explore the effect of CRGD on PEG hydrogel cell adhesion, a cell adhesion experiment was conducted on PBMSCs on the hydrogel surface (Figure 3C). As shown in the figure, the nucleus was blue, and the cytoskeleton was red. Generally, PBMSCs, as adherent cells, were fusiform in culture bottles. In the PEG-0 mM CRGD hydrogel group, fewer PBMSCs adhered to the surface of the hydrogel, and the isolated cells exhibited a round-shaped morphology. The adherent PBMSCs increased in the PEG-0.5 mm CRGD hydrogel group, but the cells were still not extended and round. In the PEG-1 mM CRGD hydrogel group, PBMSCs not only adhered to more cells, but also extended in spindle shape. The cytoskeleton of PBMSCs in the PEG-2.5 mM CRGD hydrogel group exhibited a radial morphology.

### 3.4. The Effect of Gradient Concentration of CRGD on Cell Chondrogenesis

In order to evaluate the effect of CRGD polypeptide on the chondrogenic induction of PBMSCs, an immunofluorescence assay of collagen II (COL II) was performed (Figure 4A). Compared with the PEG-0 mM CRGD hydrogel group, PEG hydrogels with 1 mM CRGD induced more cartilage differentiation in PBMSCs with the increase of CRGD concentration. However, to our surprise, PEG hydrogels with high concentrations of CRGD (2.5 mM) did not induce more chondrogenic differentiation. Semi-quantitative analysis of fluorescence intensity obtained by Image J software (Figure 4B) showed that there was no statistically significant difference in fluorescence intensity between the PEG-2.5 mM CRGD hydrogel group and the PEG-0 mM CRGD hydrogel group (*p* > 0.05). Compared with the other three groups, COL II fluorescence intensity in the PEG-1 mM CRGD hydrogel group was significantly increased (*p* < 0.05).

In addition to the comparison of cartilage differentiation at the protein level, qRT-PCR was also performed to verify the difference in cartilage differentiation at the mRNA level (Figure 4C). The results showed that COL II and ACAN in the PEG-1 mM CRGD hydrogel group were significantly higher than those in other three groups (*p* < 0.01). The expression of chondrogenic genes in the PEG-2.5 mM CRGD hydrogel group was significantly higher than that in the PEG-0 mM CRGD hydrogel group (*p* < 0.01). There were no significant differences in other groups (*p* > 0.05). These results suggest that hydrogels with an optimum concentration of CRGD (1 mM) induced a greater amount of chondrogenic differentiation than unmodified PEG hydrogels. Previous studies [21,22,34] have also shown that high concentration of continuously existing RGD could inhibit chondrogenesis, while the temporary presence of RGD could promote chondrogenesis. Moreover, the greater pore size and mechanical properties of PEG-CRGD hydrogels are reportedly beneficial for cartilage differentiation [35,36]. The poor mechanical properties of PEG-2.5 mM CRGD hydrogels might be one of the important reasons for the decrease of chondrogenic abilities. 

### 3.5. The Effect of CRGD Gradient Concentration on Immunophenotype

Raw264.7 cells were cultured in different concentrations of CRGD hydrogel for 7 days in vitro to detect the trend of M1 or M2 polarization of Raw264.7 macrophages. M1 macrophages secrete high levels of pro-inflammatory factors that promote inflammation; M2 macrophages are characterized by high expression of scavenger receptors, which perform phagocytosis and immunomodulatory functions and can promote tissue repair. According to Figure 5, CRGD can downregulate M1 markers (IL-1, INOS) and upregulate M2 markers (IL-10, Arg). There was no significant difference in M1-related genes (IL-1, INOS) between the PEG-1 mM CRGD hydrogel group and the PEG-2.5 mM CRGD hydrogel group (*p* > 0.05). Compared with PEG hydrogels without CRGD modification, the INOS gene in the PEG-1 mM CRGD hydrogel group was downregulated (*p* < 0.01), while IL-1 had no significant difference. For M2-related genes (IL-10, Arg), the M2 gene phenotype of the CRGD-modified PEG hydrogel group was significantly higher than that of the PEG hydrogel group without CRGD modification (*p* < 0.05). Interestingly, the expression of IL-10 in the PEG-2.5 mM CRGD hydrogel group was significantly higher than that in the other three groups. The expression of Arg in the PEG-0 mM CRGD hydrogel group and the PEG-0.5 mM CRGD hydrogel group was significantly lower than that in the PEG-1 mM CRGD hydrogel group (*p* < 0.05).

Inspired by the results of immunoregulation of Raw264.7 cells in hydrogel, we further evaluated the immune response of C57/BL6 mice implanted subcutaneously with PEG hydrogel modified with different concentrations of CRGD for 30 days. H&E and M&T staining results of tissues around the hydrogel showed (Figure 6A) that with the increase of CRGD concentration, the aggregation of inflammatory cells around the hydrogel increased (the position of “I” in Figure 6A indicates the thickness of infiltrated inflammatory cells around the hydrogel), which may be related to the cell adhesion of CRGD. It is worth noting that with the increase of CRGD concentration, the number of blood vessels around the hydrogel also increased (the arrow in Figure 6A indicates the representative new blood vessel tissue), suggesting that CRGD may have the function of promoting angiogenesis.

To further explore the immunomodulatory effects of PEG hydrogels modified with different concentrations of CRGD, immunohistochemical observations were performed to assess the phenotype of macrophages infiltrated into tissues adjacent to the hydrogels (Figure 6B). CD86 is the surface marker of M1 macrophages. CD206 is a surface marker of M2 macrophages. The results showed that with the increase of CRGD concentration, the infiltration of M2 macrophages (CD206^+^) around PEG-RGD hydrogel gradually increased, while the proportion of M1 (CD86^+^) macrophages decreased with the increase of CRGD concentration. (The positive cells were stained brown, and the nuclei were stained blue with hematoxylin.) Studies have shown that RGD can regulate macrophage function by binding to specific integrin sites [5]. Therefore, CRGD may play a role in the polarization of macrophages toward M2 by acting on integrins.

## 4. Conclusions

In summary, we synthesized a highly efficient and initiator-free PEG-CRGD hydrogel by the Michael addition reaction of maleimide (SH-MAL). The CRGD improved the interaction between PBMSCs and PEG hydrogel; specifically, CRGD can promote proliferation and adhesion. The 1 mM CRGD-modified PEG hydrogels had the best ability to promote chondrogenic differentiation. At excessive concentrations, CRGD-modified PEG hydrogels inhibited chondrogenic differentiation. As for immunity regulation, CRGD could regulate the polarization of macrophages to M2 phenotype to promote tissue repair. The PEG-CRGD hydrogel combined with PBMSCs has the potential to be a suitable scaffold for tissue engineering cartilage. In addition, the results may inspire the development of CRGD-based materials for not only cartilage repair, but also various other tissue engineering such as bones, tendons, corneas, and skin.

## Figures and Tables

**Figure 1 pharmaceutics-14-02622-f001:**
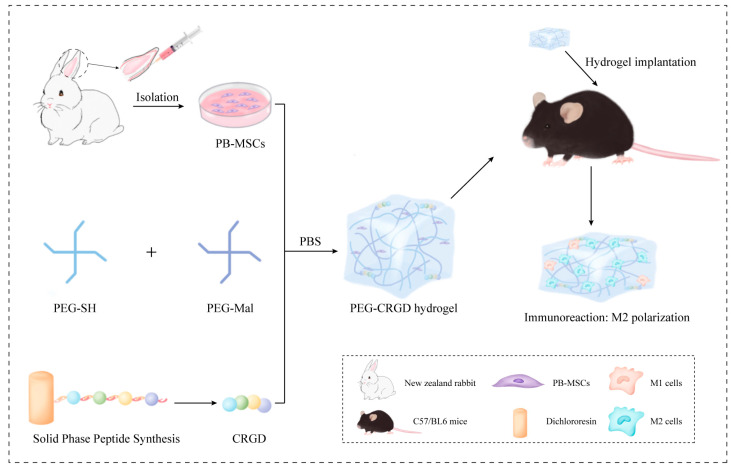
Illustration of PBMSCs in PEG-CRGD hydrogels.

**Figure 2 pharmaceutics-14-02622-f002:**
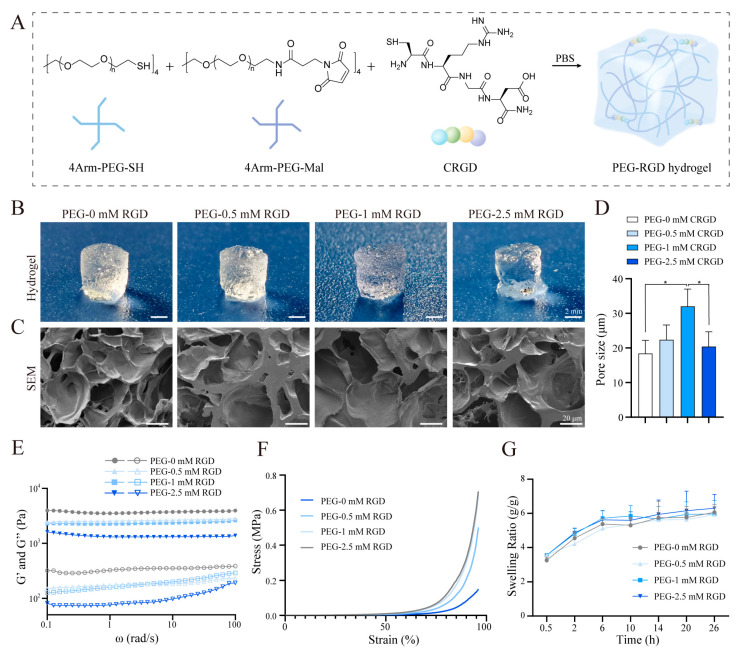
Materials characterization of PEG-CRGD hydrogels (**A**) Chemical structures of 4Arm-PEG-SH, 4Arm-PEG-Mal, and RGD schematic diagram of various cross-linked reactions in hydrogels. (**B**) General view of PEG-CRGD hydrogel scaffolds. (**C**) Scanning electron micrographs of PEG-CRGD hydrogel scaffolds. (**D**)The average pore size of PEG-CRGD hydrogels. * *p* < 0.05. (**E**) Rheological frequency sweeping of PEG-CRGD hydrogels. (**F**) Strain–stress curves of PEG-CRGD hydrogels in compression study (solid icon = storage modulus: G′; hollow icon = loss modulus: G″). (**G**) Swelling curves of PEG-CRGD hydrogels.

**Figure 3 pharmaceutics-14-02622-f003:**
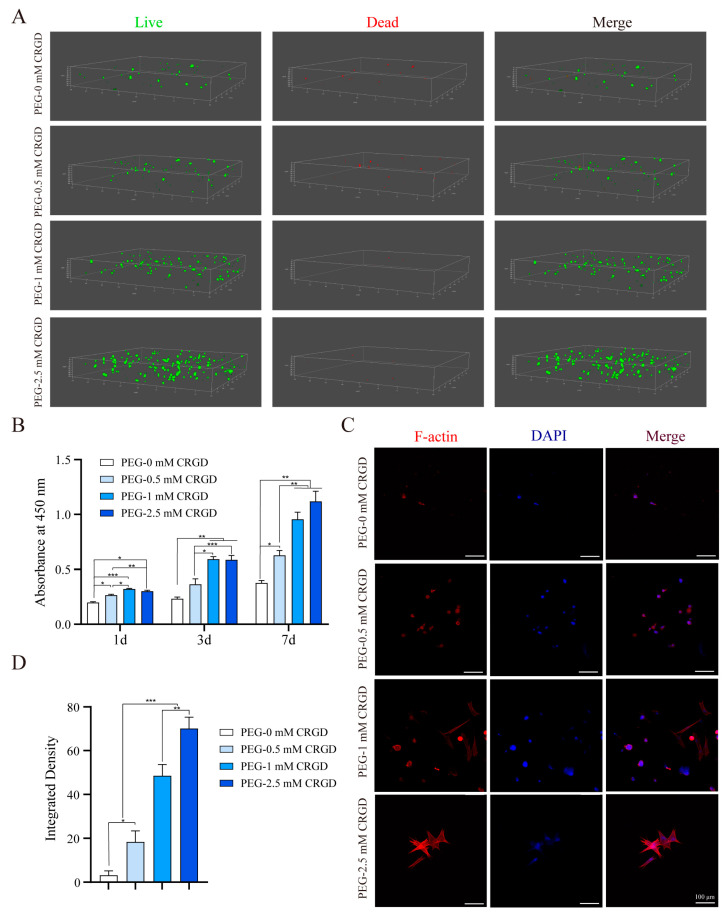
The proliferation and adhesion of PEG-CRGD hydrogels. (**A**) Live/dead staining in 3D plots of PBMSCs. (**B**) Cell proliferation determined in PEG-CRGD hydrogels by CCK8 assay. (**C**) Cell adhesion experiment on PEG-CRGD hydrogels surfaces. (**D**) Semiquantitative determination of cell adhesion experiment. * *p* < 0.05, ** *p* < 0.01, *** *p* < 0.001.

**Figure 4 pharmaceutics-14-02622-f004:**
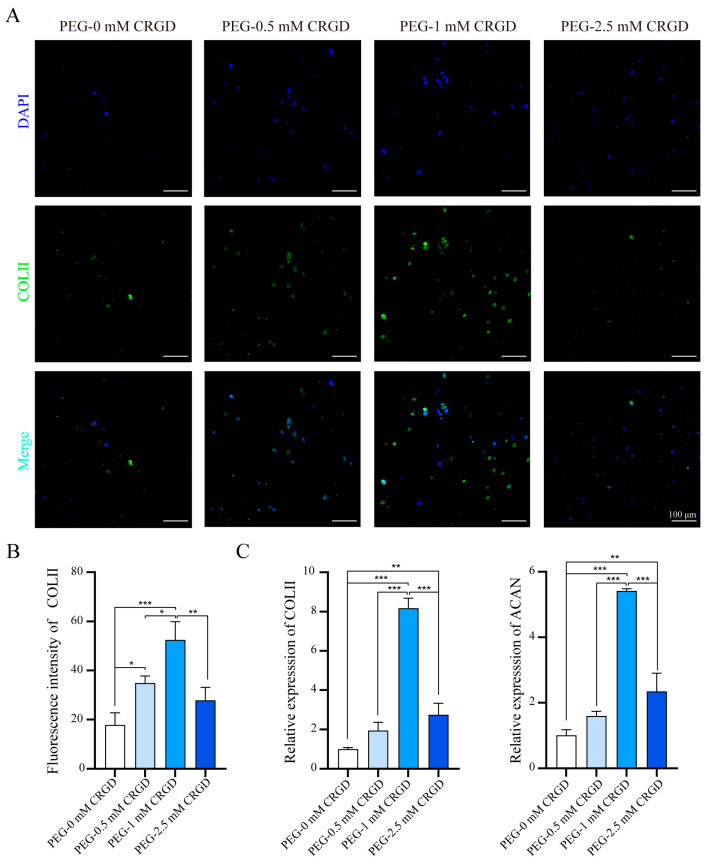
The chondrogenic differentiation of PBMSCs in PEG-CRGD hydrogels. (**A**) Immunofluorescence staining of COL II expression in PBMSCs encapsulating PEG-CRGD hydrogels. Scale bar = 100 μm. (**B**) The quantification of COL II. (**C**) The relative mRNA expression of chondrogenic genes (COL-II and ACAN) in PBMSCs. * *p* < 0.05, ** *p* < 0.01, *** *p* < 0.001.

**Figure 5 pharmaceutics-14-02622-f005:**
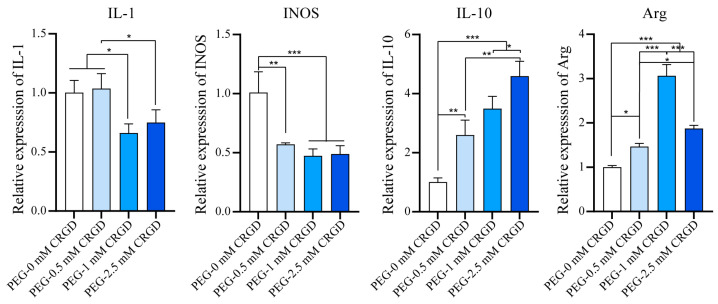
The mRNA levels of the M1(IL-1 and INOS)/M2(IL-10 and Arg) macrophages biomarkers of RAW264.7 in hydrogels. * *p* < 0.05, ** *p* < 0.01, *** *p* < 0.001.

**Figure 6 pharmaceutics-14-02622-f006:**
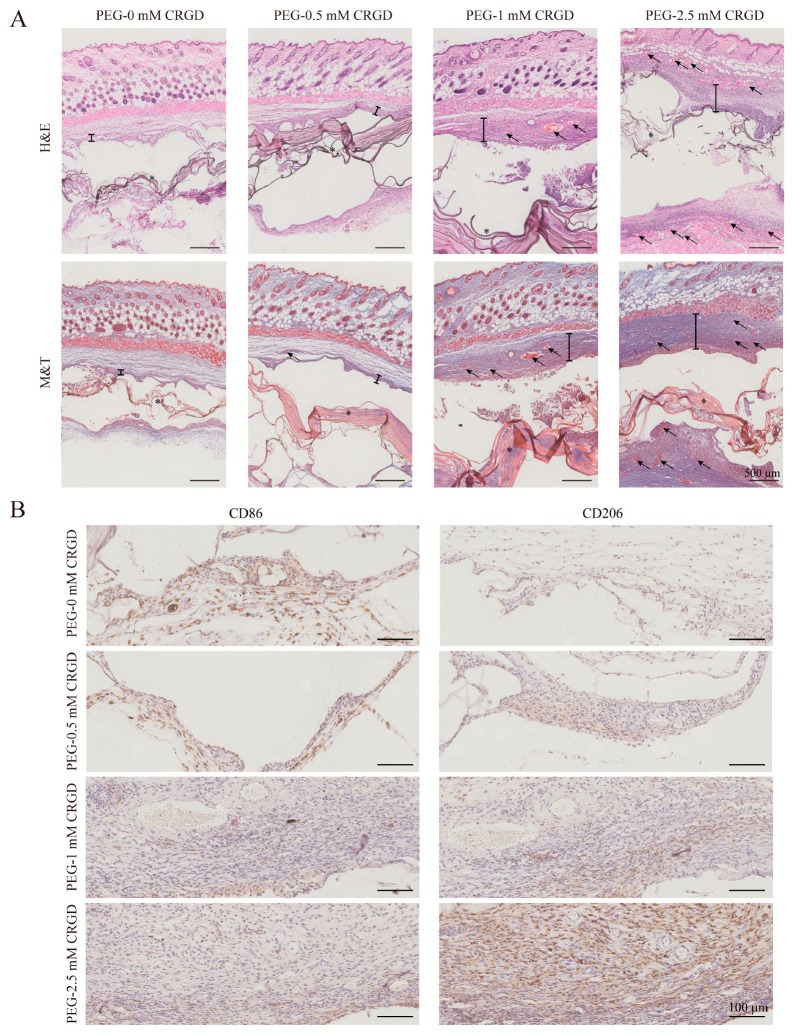
Macrophage polarization of PEG-CRGD hydrogels implanted in C57/BL6 mice. (**A**) H&E and M&T staining were used to evaluate the inflammatory response after 30 days implantation. Scale bars, 500 μm; * = hydrogels; “I” = the thickness of infiltrated inflammatory cells around the hydrogel; arrow = the representative new blood vessel tissue (**B**) Immunohistochemical staining to assess the phenotype of macrophages infiltrated into hydrogel–tissue interfaces on day 30 after implantation. Scale bars, 100 μm.

## Data Availability

Not applicable.

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
