# Peer review of "Immunomodulatory PEG-CRGD Hydrogels Promote Chondrogenic Differentiation of PBMSCs"

_pharmaceutics, 2022, doi:10.3390/pharmaceutics14122622_

Round 1

Reviewer 1 Report

The manuscript represents a promising research idea, but the following criteria need to be addressed:

1. The prepared hydrogel should be in vitro evaluated for:

1. Rheological behavior

2. Swelling studies

3. Stability testing

Author Response

Dear Editor Carina Li:

Dear Reviewer 1:

Many thanks for your comments. The comments are valuable and very helpful for improving upon our paper. And the manuscript was revised accordingly and the answers to comments were listed below.

Question 1: The prepared hydrogel should be in vitro evaluated for: 1. Rheological behavior 2. Swelling studies 3. Stability testing

Thank you very much for your comments and suggestions. It is true that our previous consideration was not comprehensive enough. We have supplemented the above experiments in our paper (Figure 2E-G).

Thank you very much for your comments, they are very helpful to improve our paper. We have carefully reviewed and addressed all comments. Looking forward to your further guidance!

Thank you for your time and consideration!

Reviewer 2 Report

The work presents a complex study of biological activity of PEG-CRGD hydrogels, the results are interesting and are recommended for publication

Minor comments

The description of the preparation procedure is unclear (lines 118 and further). Authors should first specify they used 4 hydrogel samples with different CRGD content

"0.5 mM CRGD-PEG hydrogel was prepared and mixed with 48.87 μL PEG-SH4, 50 120 μL PEG-MAL4 and 1.13 μL CRGD" is confusing since it was prepared by mixing and not prepared and mixed.

What are PEG-SH and PEG-SH4? and the same for PEG-Mal. CCK8 and CCK-8 should be also unified.

What were the parameters of SEM images scanning?

How was the pore size determined, it should be presented with standart deviation, it looks the same on the images

Hydrogels characterization should also include mechanical properties

Line 209 "The proliferation of PBMSCs in the hydrogels of the four groups was similar to that of CCK8" is unclear, what is the CCK8 proliferation?

Fig 4 (C) caption is incorrect

Reviewer 3 Report

The authors have reported the creation of hydrogels composed of chemically modified polyethylene glycols and cell adhesion peptide, CRGD, as new scaffolds for cartilage formation. These results will be helpful and informative for researchers in the field of biomaterials. Whereas the reviewer thinks that the authors’ study in this manuscript is quite interesting, suggestive, and well-organized, some descriptions are not enough. The authors’ manuscript is not suitable for publication in “Pharmaceutics” in the present form.

From these considerations, the reviewer recommends to accepting for publication in “Pharmaceutics,” if the following issues are resolved.

1)      There is no characterization or physical properties of the new hydrogels. How soft or hard are these hydrogels?

2)      There are comparisons between hydrogels prepared by the authors but no comparisons with hydrogels from other researchers. Why don't the authors make comparisons with other studies?

3)      Why did the authors' various experiments with "1M CRGD" hydrogels show better results? These results need to be discussed.

4)      Page 3, line 106 (2.2. Preparation of Hydrogel): “dichlororesin” should be detailed.

5)      Page 3: DMF, DIEA, and DTT should be explained.

6)      Page 3, line 119, "50 µL PEG-SH4:50 µL PEG-Mal4": What are “4s”

Reviewer 4 Report

The manuscript (pharmaceutics-2039450) entitled "Immunomodulatory PEG-CRGD hydrogels promote chondrogenic differentiation of PBMSCs" needs to be revised considering the following suggestions: 

1.  A illustration should be included in the introduction section which highlights the hypothesis behind the current investigation and its drug delivery perspectives utilizing the hydrogel system.

2.  Caption Figure 1 should be modified as "Illustration of PBMSCs in PEG-CRGD hydrogels". Delete the word "cartoon" from the present caption of Figure 1.  

3. Delivery perspectives of the developed system for in vivo application should be included in the conclusion section.  
